# Technological Developments, Exercise Training Programs, and Clinical Outcomes in Cardiac Telerehabilitation in the Last Ten Years: A Systematic Review

**DOI:** 10.3390/healthcare12151534

**Published:** 2024-08-02

**Authors:** Marina Garofano, Carmine Vecchione, Mariaconsiglia Calabrese, Maria Rosaria Rusciano, Valeria Visco, Giovanni Granata, Albino Carrizzo, Gennaro Galasso, Placido Bramanti, Francesco Corallo, Carmine Izzo, Michele Ciccarelli, Alessia Bramanti

**Affiliations:** 1Department of Medicine, Surgery and Dentistry, University of Cagliari, 09124 Cagliari, Italy; 2Department of Medicine, Surgery and Dentistry, University of Salerno, 84084 Fisciano, Italy; cvecchione@unisa.it (C.V.); macalabrese@unisa.it (M.C.); mrusciano@unisa.it (M.R.R.); vvisco@unisa.it (V.V.); ggranata@unisa.it (G.G.); acarrizzo@unisa.it (A.C.); ggalasso@unisa.it (G.G.); cizzo@unisa.it (C.I.); mciccarelli@unisa.it (M.C.); abramanti@unisa.it (A.B.); 3Faculty of Psychology, University eCampus, 22060 Novedrate, Italy; bramanti.dino@gmail.com; 4Centro Neurolesi Bonino Pulejo, 98124 Messina, Italy; francesco.corallo@irccsme.it

**Keywords:** cardiac telerehabilitation, cardiac remote rehabilitation, coronary artery disease, CAD, CTR

## Abstract

Background: Cardiovascular diseases (CVDs) are associated with very high rates of re-hospitalization and mortality worldwide, so the complexity of these pathologies requires frequent access to hospital facilities. The guidelines also emphasize the importance of cardiac rehabilitation (CR) programs, which have demonstrated a favorable effect on outcomes, and cardiac telerehabilitation (CTR) could represent an innovative healthcare delivery model. The aim of our review is to study how technologies used in rehabilitation have changed over time and also to understand what types of rehabilitation programs have been used in telerehabilitation. Methods: We searched randomized controlled trials (RCTs) in three electronic databases, PubMed, Web of Science, and Scopus, from January 2015 to January 2024, using relevant keywords. Initially, 502 articles were found, and 79 duplicates were identified and eliminated with EndNote. Results: In total, 16 RCTs fulfilled the pre-defined criteria, which were analyzed in our systematic review. The results showed that after CTR, there was a significant improvement in main outcome measures, as well as in relation to technological advances. Conclusions: Moreover, compared to center-based rehabilitation, CTR can offer further advantages, with better cost-effectiveness, the breakdown of geographical barriers, and the improvement of access to treatment for the female population, which is traditionally more socially committed.

## 1. Introduction

Among chronic diseases, cardiovascular diseases (CVDs) are associated with very high rates of re-hospitalization and mortality worldwide. The complexity of these pathologies requires frequent access to hospital facilities in order to allow multidisciplinary assessments and for the execution of diagnostic tests, resulting in increased direct and indirect costs (loss of work days, travel, inappropriate hospitalizations) [1]. Moreover, technological evolution and the experiences gained during the COVID-19 emergency have shown the need to develop new organizational models based on the management of patients with CVD in telemedicine. The use of these technologies allows the patient to be taken in and supervised at home, allowing for a rehabilitation program to be conducted remotely and safely.

In order to reduce hospitalizations and improve the prognosis and the quality of life of CVD patients, especially with heart failure (HF), the recent guidelines [2,3] indicate the need for drug therapy based on four pharmacological “pillars” (Angiotensin-Converting Enzyme inhibitors—ACE inhibitors; Angiotensin Receptor Blocker—ARB; Angiotensin Receptor–Neprilysin Inhibitor—ARNI; mineralocorticoid receptor antagonists—MRA; beta blocker; Sodium–Glucose Cotransporter 2—SGLT2 inhibitors).

However, the international guidelines also emphasize the importance of cardiac rehabilitation (CR), which is an active and dynamic multifactorial process that includes exercise training, cardiac risk factor modification, psychosocial assessment, and outcome assessment [4], which have demonstrated favorable effects on clinical stability, reducing disability and the risk of subsequent cardiovascular events, supporting the maintenance and resumption of an active role in society, and improving survival and quality of life [5]. Consequently, CR and supervised training may be considered the fifth pillar in HF treatment [6], and the beneficial effects are evident in patients who adhere to long-term CR programs [7,8]. Unfortunately, despite much evidence, supervised training and CR programs are underestimated and underutilized [9], especially due to geographical social barriers and clinical conditions. Long-term participation and adherence to these programs should, therefore, be encouraged and strengthened. Telerehabilitation and mHealth can help to achieve this goal, as suggested in the international guidelines [10]. Telerehabilitation is a branch of telemedicine [11] in which information and communication technologies (ICTs) and, in advanced cases, remote-control technologies such as robotics are used to directly provide remote rehabilitation activities [12,13]. These technologies also include messaging and calls via telephone and/or the Internet, video conferencing communications between the patient and the healthcare provider, and sensor and telemonitoring systems [1]. In the case of telerehabilitation, the systems that allow information to be transmitted are as follows [14,15]:Hardware and software systems that acquire and process signals, images, and data;Applications that permit the transmission of health information in a bidirectional way;Dedicated web portals or information technology platforms.

In most cases, telerehabilitation is carried out by providing the patient with videos in which the exercises to be performed are shown. These exercises then can be performed either by videoconference with the therapist (“real-time” telerehabilitation, in which both the healthcare provider and the patient are online) or without supervision (in this case, however, the healthcare provider is “off-line”). In some cases, kits containing motion sensors, tablets with customized training software, and other peripherals can be provided to patients [16]. Telemedicine and all its components, including telerehabilitation, are recognized as relevant both nationally and internationally, so they are considered a “cultural revolution” [17]. This cultural revolution, however, is necessary because of the aging population and the resulting increased incidence of chronic diseases [18]. All of the above-mentioned indicate the need for innovative healthcare delivery models that support more patient-centered care in order to meet health needs more effectively, reducing current geographic gaps, ensuring a better “experience of care,” and improving the efficiency levels of healthcare systems through the promotion of home care and remote monitoring protocols [19]. Consequently, the aim of our review is to study how technologies used in rehabilitation have changed over time and also to understand what types of rehabilitation programs have been used in telerehabilitation.

## 2. Materials and Methods

### 2.1. Study Design

This systematic review was performed according to the Preferred Reporting Items for Systematic Review and Meta-Analyses (PRISMA) statement [20].

### 2.2. Literature Search

A literature analysis was performed to identify relevant studies in 3 electronic databases, PubMed, Web of Science (WOS), and Scopus, systematically searched from January 2015 to January 2024, taking into account only articles published in peer-reviewed journals, with the following keywords: (Cardiovascular Disease OR Disease, Cardiovascular OR Major Adverse Cardiac Events OR Cardiac Events OR Cardiac Event OR Event, Cardiac OR Adverse Cardiac Event OR Adverse Cardiac Events OR Cardiac Event, Adverse OR Cardiac Events, Adverse) AND (Telerehabilitation OR Tele-rehabilitation OR Tele rehabilitation OR Tele-rehabilitations OR Remote Rehabilitation). Complete search strategies are provided in Appendix A.

### 2.3. Inclusion and Criteria

In this review, we included articles published from January 2015 to January 2024 that were written in English language and had available full texts, along with studies targeting only humans that focused on telerehabilitation of patients with cardiovascular disease. We also defined and applied the following inclusion criteria to identify eligible papers: randomized controlled trials (RCTs) in which adults (≥18 years old) with CVD (i.e., myocardial infarction, angina, heart failure, or after revascularization) attend a cardiac telerehabilitation (CTR) program in comparison with a control group attending usual-care or center-based CR.

### 2.4. Study Selection

Before starting the article selection phase, duplicate records were removed with EndNote. After that, all articles were screened for title and abstract by two independent reviewers who separately selected and discussed conflicts about doubtful cases regarding the inclusion/exclusion criteria. Next, the review authors conducted full-text screening based on inclusion and exclusion criteria to determine the papers’ eligibility. During this process, if necessary, a third author resolved potential disagreements via discussion.

### 2.5. Risk of Bias and Quality Assessment of Studies

All the studies included are evaluated using the Cochrane risk-of-bias tool to assess the risk of all types of bias (selection bias, performance bias, attrition bias, reporting bias, and overall bias) [21] (Table 1). After that, all the studies were also checked for the Physiotherapy Evidence Database (PEDro) score to perform a quality assessment. The PEDro score [22] comprises 11 items, with a maximum of 10 points, because the first item (eligibility criteria) does not contribute to total score. Studies with a PEDro score of 9–10 points are considered excellent, studies with a score of 6–8 points are considered good, and, finally, studies with a score of ≤5 are considered poor (Table 2). Among the included studies, 10 are considered poor with a score of 4 or 5; the remaining 6 studies are considered good with a score between 6 and 8. None of the included studies are considered excellent.

### 2.6. Data Extraction

Two of the reviewers (MG, AB) independently investigated the titles and abstracts extracted from the database searches to determine if they fit the inclusion criteria. Disagreements regarding the inclusion or exclusion of a particular manuscript based on the appraisal of its abstract were determined by reaching an agreement or consulting an additional reviewer (MC). Data extraction arrangements were established based on the current literature in the field and the research questions. Extraction of the data was based on essential information according to questions of the current review, such as (a) author, country, and year; (b) study design; (c) population; (d) diagnosis; (e) telerehabilitation; (f) control; (g) technological solutions; (h) outcomes; and (i) results (Table 3). In addition, for quicker reference, Table 4 shows the results of the studies and the qualitative assessment with the PEDro scale.

## 3. Results

### 3.1. Study Selection and Characteristics

Initially, 502 articles were found through database searches on PubMed, WOS, and Scopus; after that, 79 duplicates were identified and eliminated with EndNote. A total of 423 articles were screened for titles and abstracts; after that, 76 papers were selected to be potentially relevant for the present review. Finally, 16 articles were included after full-text screening. This selection process is summarized in the PRISMA flow diagram (Figure 1), and in Table 3, there are the descriptive characteristics of the 16 studies included in the present review, with a focus on the study design, population, diagnosis, intervention, technological solution, measured outcomes, and results. Regarding the diseases assessed, twelve studies are about coronary artery diseases (CAD), one concerned patients with coexisting chronic obstructive pulmonary disease (COPD) and chronic heart failure (CHF), one concerned patients who underwent ablation for atrial fibrillation, and two concerned patients with HF.

### 3.2. Monitoring Devices

The technological solutions adopted are the most varied and provide for the detection of vital parameters, which are transmitted in real-time by the device that detects them [24,25,27,33,35] or uploaded later by the patients using applications [23,26,34] in order to allow the safety of the patients during training and the monitoring of the correctness of the rehabilitation program. Some studies did not involve the detection of vital parameters but only the correct execution of movement by a sensor or with the use of virtual reality in CR [28,37,38]. Studies based on virtual reality do not generally aim to improve exercise tolerance but are aimed at anxiety and depression management [29,30]. In contrast, the study by Lima et al. [31], which also includes the use of virtual reality, is aimed at improving lung function in post-coronary artery bypass graft (CABG) patients. In the study conducted by Snoek et al. [36], the devices used are able to detect and transmit vital and exercise parameters in real time. Finally, in the 2015 study by Maddison et al., [32], there is no monitoring system; all outcome measures are based on patient’s self-reported data. In a 2020 study, on the other hand, the same authors use a very comprehensive system that can directly detect and transmit vital and movement parameters [33].

### 3.3. Intervention

As regards CR programs, the studies included in our review show a wide range of interventions, based mainly on exercise, sometimes accompanied by patient education on the management of risk factors (dietary interventions, cessation of smoking habits, maintenance of an active lifestyle) and counseling interventions for the management of anxiety and depression. These studies also show a wide range of rehabilitation programs in terms of types, intensity, and duration of exercise training. The rehabilitation programs mostly focus on exercise and range from a minimum of 6 weeks with a frequency of three times per week [27] to a maximum of 24 weeks [32,36,37,38] with a frequency ranging from three to five times per week.

In these studies, the exercise type is moderate-intensity aerobic training [32,36] alone or combined with strength training and stretching [37,38]. Three studies involve an 8-week home-based exercise program [25,34,35], which, in the case of Piotrowicz et al., 2020 [35], follows an initial 1-week center-based training. In all the above-mentioned studies, combined training, consisting of interval training exercises for strength training with TheraBand or calisthenics in addition to breathing exercises and endurance training (with an intensity ranging from 40 to 80% of the heart rate reserve (HRR)), is used.

Three studies provide a treatment duration of 12 weeks [23,26,33] with a frequency of at least three times or 150 min per week and an indication to be active at least 5 days per week. The exercise programs reported in these papers are based on endurance training at an intensity ranging from 40 to 80% of HRR. Only one study provides a personally tailored 16-week exercise program [24] with an intervention based on both endurance and resistance training, together with patient education about the desirability of maintaining a healthy lifestyle and practicing daily physical activity.

With regard to the studies including the use of virtual reality [29,30,31] associated with exercise training, there are generally only a few sessions, ranging from four to eight, aimed to improve symptoms such as anxiety and depression [29,30], which are often present in people with cardiovascular disease, or to enhance the respiratory pattern in post-CABG patients [31]. Despite the wide variety of rehabilitation programs proposed, we report the lack of studies involving exercises to improve flexibility and coordination, and only a few studies have included respiratory exercises.

### 3.4. Primary and Secondary Outcomes

As the primary outcome, most of the included studies consider cardiorespiratory fitness (CRF) assessed with cardiopulmonary exercise testing (CPET), expressed as peak oxygen consumption (VO2 peak) [23,26,32,34,36], maximal oxygen consumption (VO2 max) [33] or as metabolic equivalent of task (MET) [25]. Only two studies performed a 6-Minute Walking Test (6MWT) to evaluate the possible effects of CTR on exercise capacity [24,27]. The study by Frederix et al. is the only one that performs a cost analysis of CTR compared to center-based rehabilitation [28], while the two studies by Jozwik et al. take into account anxiety and depression levels assessed with the Hospital Anxiety and Depression Scale (HADS) [29,30].

As secondary outcomes in the studies analyzed, we mostly find assessments of physical activity, quality of life (QoL), training adherence, cardiovascular risk factors/laboratory parameters, and anxiety and depression levels. Physical activity is assessed in different ways, i.e., with the Physical Activity Scale for Elderly (PASE) [24], with the International Physical Activity Questionnaire (IPAQ) for the self-reported PA assessment [26,32,36], and finally, with the use of accelerometry data [28,33].

In several studies, QoL is also assessed, mainly with the Short Form 36 (SF-36) questionnaire [23,25,27,34,35]; two studies instead used the EuroQol five-dimensional (EQ5D) questionnaire for the evaluation of the QoL [32,33], whilst one study combined it with the Minnesota Living with Heart Failure Questionnaire (MLHFQ) [24].

Four studies also report an evaluation of treatment adherence [23,26,33,34] calculated according to the records provided by wearable devices like heart rate zones, accelerometers, and electrocardiogram (ECG) recording devices or through the compilation of exercise diaries.

The evaluations of cardiovascular risk factors/laboratory parameters like the anthropometric measures (body mass index, waist and hip circumferences), biochemical parameters (glucose, total cholesterol, low-density lipoprotein (LDL), high-density lipoprotein (HDL), cholesterol, triglycerides, glycated hemoglobin—HbA1c—and blood pressure) are reported in several studies included [25,33,36,37].

Finally, a psychological status evaluation was conducted using the Hospital Anxiety and Depression Scale (HADS) and the Patient Health Questionnaire (PHQ) [29,30,36]. Cai et al. also used the Health Beliefs Related to Cardiovascular Disease Scale and the Exercise Self-Efficacy Scale to evaluate the effects of CTR treatment on their patients.

## 4. Discussion

The aim of this review is to highlight how much telerehabilitation has changed since its beginnings. The technologies used in the included studies show a considerable evolution over time from simpler systems, such as text messages sent via telephone, to more sophisticated platforms, which are also equipped with virtual reality. In the studies dated between 2015 and 2018 [24,28,32], in fact, telerehabilitation was mostly designed as text messages, video messages, e-mails, and phone calls made by physiotherapists or nurses aimed to increase the exercise habits of the study participants; even in the 2015 study by Maddison et al., no patient monitoring system was provided [32].

In a 2019 paper published by the same authors [33], however, we also note a strong evolution of the telerehabilitation concept. In this study, in fact, telerehabilitation was delivered via the REMOTE-CR platform, one of the first platforms for telerehabilitation delivery, which was associated with a smartphone and sensors for the detection of vital parameters and movement that transmitted information in real time. Among the older studies, there are two [25,34] that already showed a different concept of telerehabilitation, very similar to how we understand it today, characterized, therefore, by personalized exercise programs monitored via devices integrated with web apps.

In addition to remote monitoring systems and platforms, there are also studies involving the use of virtual reality systems in CR with the aim of treating anxiety and depression and improving breathing quality [29,30,31]. Virtual reality in the management of such disorders is actually emerging as a very effective intervention, as it provides users with safe, non-threatening environments where patients can experience a different world and learn how to cope with their anxiety and depression [39]. These results are in line with the digitalization process in healthcare and are very encouraging as they go in the direction of overcoming the concept that technologies in rehabilitation deprive the patient of contact with the physiotherapist, on which the therapeutic relationship has always been based, and can therefore constitute a new and equally effective way of treating the patient as a whole [12,40,41].

Another aim of this review is to show what type of rehabilitation program have been mainly used in telerehabilitation and to highlight the effectiveness of telerehabilitation in the treatment of CVD. The rehabilitation treatments used in the included studies are heterogeneous in terms of program duration, single treatment duration, intensity, and type of exercise. In most of the included studies, in line with previous systematic reviews [8,42,43,44,45], telerehabilitation is able to improve the main outcome measures despite the great variety of the proposed interventions. In the included studies, the treatment designed for the control group is often not described in detail and is rarely supervised by a physiotherapist. On the other hand, when the control group carries out a supervised exercise program, which overlaps with the study group, the results achieved are similar [23,25,35,36] or even lower [32] for the telerehabilitation group.

Most of these studies, however, involve the recruitment of a small sample, like Batalik et al. [23] and Bravo-Escobar et al. [25], so in order to better understand the effects of telerehabilitation, the number of patients recruited should be enlarged, and the exercise programs carried out should be detailed, perhaps even tried to standardize them in relation to pathologies.

However, this contrasts with a study of 850 patients with HF, in which a 9-week full hybrid telerehabilitation program improved pVO2 and QoL but did not increase the percentage of days alive and out of the hospital or reduce mortality and hospitalization over a follow-up period of 14 to 26 months [35]. In the study by Maddison et al. 2015 [32], telerehabilitation was ineffective in improving outcomes. Despite recruiting a sample of 171 participants, this study was based on self-reported outcome measures and also did not include monitoring systems, so the own perceptions of training intensity were likely to be lower than the true physiological intensity required to impact exercise capacity.

In the remaining included studies, however, telerehabilitation proved to be an effective and safe intervention in improving patients’ CRF expressed as pVO2 [26,34], VO2 max [33], or difference in the meters walked at the 6MWT [24,27]; also, these used together with virtual reality is effective in improving anxiety and depression levels [29,30], selective attention, and conflict resolution ability [38] and in reducing risk factors (i.e., waist-to-hip ratio, BMI, ingestion of total fat) [37], with a better cost-effectiveness ratio than center-based cardiac rehabilitation alone [28]. These results are highly encouraging for the implementation of telerehabilitation and telemonitoring systems, especially thanks to the creation of devices that are increasingly easy to use and reliable, which allow rehabilitation to be carried out safely, overcoming geographical and socio-economic gaps. In order to make these technologies more effective and widely used, however, a new approach is needed from both practitioners’ and patients’ sides. This requires extensive digital literacy and the ability to transfer the traditional therapeutic relationship into a new setting that certainly offers promising possibilities for interaction, as demonstrated by the high patient compliance and adherence to telerehabilitation [46,47], which, however, should be further studied and deepened in the various fields of application [48,49,50].

## 5. Conclusions

With this systematic review, we have shown, overall, that CTR can be an advantageous alternative in improving the functional outcome of patients with CVD, especially due to the technological advances we have been assisting in recent years, which allow real-time monitoring and transmission of vital and movement parameters, offering a care experience comparable to traditional center-based rehabilitation. Moreover, compared to center-based rehabilitation, CTR can offer further advantages, with better cost-effectiveness, the breakdown of geographical barriers, and the improvement of access to treatment for the female population, which is traditionally more socially committed. Furthermore, CTR treatment is safe, can lead to increased levels of participation, and can improve long-term cardiovascular risk management. We also hope that international guidelines will be produced with the aim of reducing the variety of treatment programs and thus improving their effectiveness, and we hope that more studies will be conducted to investigate the long-term benefits of telerehabilitation.

## 6. Limitation

This systematic review has several limitations, such as small sample size and composition—in fact, many studies had fewer than 100 participants, and most of the patients were males. In addition, socio-demographic data, cultural and economic level, and place of residence are often lacking. However, such information would be necessary to truly understand the effectiveness and safety of telerehabilitation in the treatment of CVD. Additionally, the included studies were limited to papers written in English papers, and original published research articles were only searched for in three databases.

## Figures and Tables

**Figure 1 healthcare-12-01534-f001:**
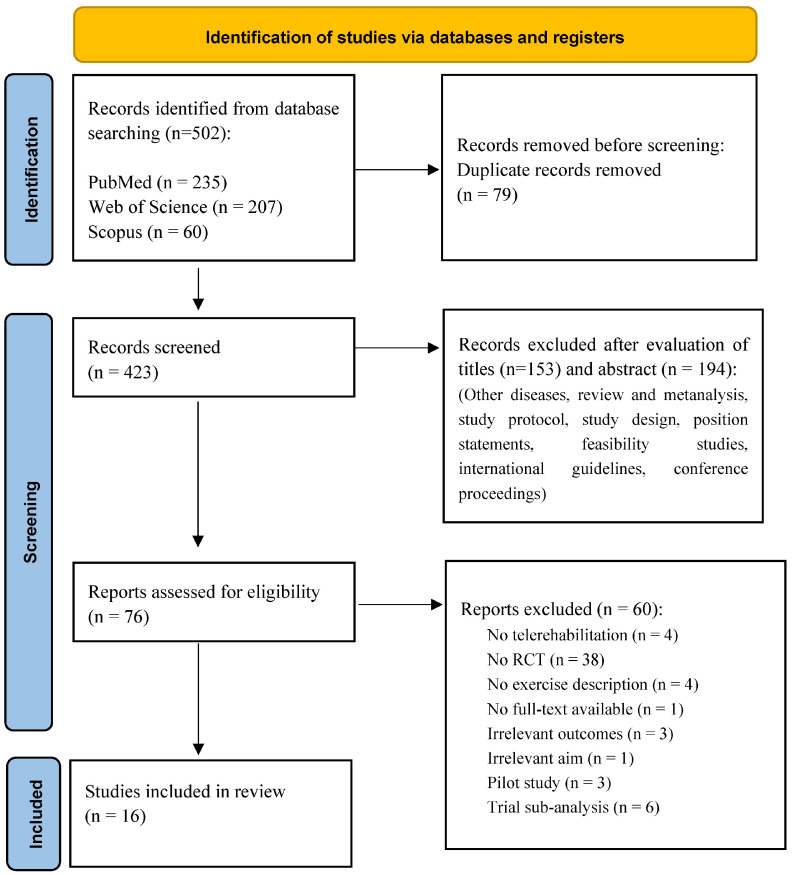
Flow diagram of study selection.

**Table 1 healthcare-12-01534-t001:** Cochrane risk of bias tool for the risk of bias in individual studies.

	Batalik et al., 2020 [23]	Bernocchi et al., 2018 [24]	Bravo-Escobar et al., 2017 [25]	Cai et al., 2022 [26]	Fang et al., 2019 [27]	Frederix et al., 2016 [28]	Józwik et al., 2021 [29]	Józwik et al., 2021 [30]	Lima et al., 2020 [31]	Maddison et al., 2015 [32]	Maddison et al., 2019 [33]	Piotrowicz et al., 2015 [34]	Piotrowicz et al., 2020 [35]	Snoek et al., 2021 [36]	Vieira et al., 2017 [37]	Vieira et al., 2018 [38]
Random sequence generation(selection bias)																
Allocation concealment(selection bias)																
Blinding of participants and personnel (performance bias)																
Blinding of outcome assessment(detection bias)																
Incomplete outcome data(attrition bias)																
Selective reporting bias(reporting bias)																
Overall																
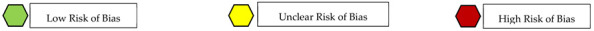

**Table 2 healthcare-12-01534-t002:** Summary of PEDro scores for included studies. 1: eligibility criteria; 2: random allocation; 3: concealed allocation; 4: similarity at baseline; 5: subject blinding; 6: therapist blinding; 7: assessor blinding; 8: completeness of follow-up; 9: intention-to-treat analysis; 10: between-group statistical comparisons; 11: point measures and variability.

	1	2	3	4	5	6	7	8	9	10	11	TOTAL SCORE	QUALITY
Batalik et al., 2020 [23]	Y	Y	N	Y	N	N	N	Y	N	Y	Y	5	POOR
Bernocchi et al., 2018 [24]	Y	Y	N	Y	N	N	Y	N	N	Y	Y	6	GOOD
Bravo-Escobar et al., 2017 [25]	Y	Y	N	Y	N	N	N	Y	N	Y	Y	5	POOR
Cai et al., 2022 [26]	Y	Y	N	Y	N	N	Y	Y	N	Y	Y	6	POOR
Fang et al., 2019 [27]	Y	Y	N	Y	N	N	N	N	N	Y	Y	4	POOR
Frederix et al., 2016 [28]	Y	Y	N	Y	N	N	Y	Y	Y	Y	Y	7	GOOD
Józwik et al., 2021 [29]	Y	Y	N	Y	N	N	Y	N	N	Y	Y	5	POOR
Józwik et al., 2021 [30]	Y	Y	N	Y	N	N	Y	N	N	Y	Y	5	POOR
Lima et al., 2020 [31]	Y	Y	N	Y	N	N	Y	N	N	Y	Y	5	POOR
Maddison et al., 2015 [32]	Y	Y	Y	Y	N	N	Y	Y	Y	Y	Y	8	GOOD
Maddison et al., 2019 [33]	Y	Y	Y	Y	N	N	Y	Y	Y	Y	Y	8	GOOD
Piotrowicz et al., 2015 [34]	Y	Y	N	Y	N	N	N	Y	N	Y	Y	5	POOR
Piotrowicz et al., 2020 [35]	Y	Y	Y	Y	N	N	Y	Y	Y	Y	Y	8	GOOD
Snoek et al., 2021 [36]	Y	Y	Y	N	N	N	N	Y	Y	Y	Y	6	GOOD
Vieira et al., 2017 [37]	Y	Y	N	Y	N	N	N	Y	N	Y	Y	5	POOR
Vieira et al., 2018 [38]	Y	Y	Y	Y	N	N	N	N	N	Y	Y	5	POOR

Note: Eligibility criteria item does not contribute to total score.

**Table 3 healthcare-12-01534-t003:** Descriptive characteristics of the included studies.

Author, Country, Year	Study Design	Population, n	Diagnosis	Telerehabilitation	Control	Technological Solution	Outcomes	Results
Batalik et al., Czech Republic, 2020 [23]	2-arm RCT	IG: n = 28 CG: n = 28	CAD (angina pectoris or myocardial infarction in the last 6 months, with left ventricular ejection fraction > 45%)	IG: **12 weeks** of training using a heart wrist monitor. Frequency: 3 days/week. Intensity: 70–80% HRR. Time: 60 min/session. Type: walking or cycling, aerobic training. These patients received feedback once a week, reflecting data uploaded on the internet application.	CG: **12 weeks** of training supervised by a physiotherapist. Frequency: 3 days/week. Intensity: Heart rate set at 70–80% heart rate reserve. Time: 60 min/session. Type: Walking or cycling and aerobic training.	Using a wrist heart rate monitor, M430 (Polar, Kempele, Finland), and monitoring heart rate, time, training mode, duration, and distance of training physical activity. Patients then upload to a Polar Flow web application secured by individual login and password.	Primary: Physical fitness with CPET parameter pVO2. Secondary: HRQoL with SF36, training adherence (percentage counted from the total number of accomplished training sessions).	No serious adverse events were recorded. Physical fitness showed significant improvement in IG and CG without significant between-group differences. The training adherence between groups was similar.
Bernocchi et al., Italy, 2018 [24]	2-arm RCT	IG: n = 56; CG: n = 56	COPD and CHF	IG: **16 weeks** of educational/exercise intervention from an NT and a PT. PT designed a personalized exercise program for each patient who was provided with mini-ergometer, pedometer, and diary. **Basic level:** 15–25 min of exercise with mini-ergometer + 30 min of callisthenic exercises 3 times/week + free walking 2 times/week. **High level:** 30–45 min of mini-ergometer with incremental load (from 0 to 60 W) + 30–40 min of muscle reinforcement exercises using 0.5 kg weights + pedometer-based walking performed between 3 and 7 days/week.	CG: **16 weeks** of standard care program including medications + educational session about the desirability of maintaining a healthy lifestyle (practice daily physical activity).	Patients were provided with a pulse oximeter (GIMA, Milan, Italy) and a portable one-lead electrocardiograph (Card Guard Scientific Survival Ltd., Rehovot, Israel) for real-time telemonitoring of vital signs. NT made a weekly structured phone call, collecting information about the disease status and symptoms and offering advice regarding diet, lifestyle, and medications. PT made a weekly phone call, verifying and planning the rehabilitation targets for the following week.	Primary: Exercise capacity with difference (Δ) in the meters walked at the 6MWT. Secondary: Time to event (hospitalization and death), dyspnea with MRC, physical activity profile with PASE, impairment/disability with BARTHEL, QoL with MLHFQ, and CAT.	Primary and secondary outcomes were significantly improved in the IG compared with the CG. IG maintained the benefits acquired at 6 months follow-up.
Bravo-Escobar et al., Spain, 2017 [25]	2-arm RCT	IG: n = 14 CG: n = 14	Stable CAD at moderate cardiovascular risk	IG: For **8 weeks**, patients went to CR unit once a week and exercised at home. Frequency: 5 to 7 days per week. Intensity: 70% of HRR (first month) and 80% of HRR (second month). Time: 60 min/session—15 min of warm-up (stretching and isotonic exercises) + 30 min of continuous aerobic exercise (alternating between treadmill and stationary bike) + 15 min cooldown. Once a week, patients performed strength training sessions consisting of 1/2 series of 10 repetitions (50% of 1MR) and attended health education sessions and group psychotherapy support.	GC: For **8 weeks**, 3 times a week of traditional cardiac rehabilitation in hospital with the same exercise program prescribed for IG.	Remote electrocardiographic monitoring device NUUBO that uses Bluetooth wireless technology and biometric vests using textile electrodes. On the first day, the patients were instructed on how to use this device’s mobile application, and they were all given a smartphone with Internet connection and the NUUBO application pre-installed.	Anthropometric measurements, blood pressure, exercise capacity (with an exertion test to ensure that they could tolerate the exercise and to determine their METs and exercise time), laboratory parameters (total cholesterol, HDL cholesterol, LDL cholesterol, triglycerides, blood glucose and HbA1c), and QoL with SF-36.	No significant differences were observed between IG and CG in terms of exercise time and METS achieved during the exertion test and the recovery rate in the first minute. QoL was higher in the CG. No serious heart-related complications were recorded.
Cai et al., China, 2022 [26]	2-arm RCT	IG: *n* = 50 CG: *n* = 50	Patients who underwent ablation for atrial fibrillation	IG: **12 weeks** of standard rehabilitative care. Time: 10 min warm-up to prevent exercise-induced injuries + 45 min at heart rate target + 10 min relax and stretch period. Frequency: 150 min every week. In addition, there was a home-based, patient-tailored, mobile application-guided, and portable electrocardiogram recording device-monitored telerehabilitation program.	CG: **12 weeks** of standard rehabilitation treatment. Time: 10 min warm-up to prevent exercise-induced injuries + 45 min at heart rate target + 10 min relax and stretch period. Frequency: 150 min every week. Adherence was determined based on patients’ exercise diaries.	Mobile application-guided and portable electrocardiogram recording device-monitored telerehabilitation program. All patients were instructed to install the ShuKang™ (Recovery Plus Inc., Chengdu, China) application on their smartphones. To support patient adherence, the app Binglijia™ (Xingshulin Inc., Beijing, China) was introduced to enable the subjects to communicate with physicians and upload and share their daily exercise training records.	Primary: Physical fitness with CPET parameter pVO2. Secondary: Adherence (percentage of the 12 weeks during which the patient completed 150 or more minutes of exercise), physical activity with IPAQ, beliefs related to cardiovascular disease, and exercise self-efficacy (with the Exercise Self-Efficacy Scale).	Patients in the IG showed improvements in VO2 peak, adherence, self-reported beliefs related to cardiovascular disease, and exercise self-efficacy.
Fang et al., China, 2019 [27]	2-arm RCT	IG: n = 40 CG: n = 40	Patients with CAD post-PCI	IG: **6-week** program, including self-education and exercise program. Type: Outdoor walking or jogging with real-time physiological monitoring. Frequency: At least thrice/week. They also received two home visits by a physical therapist to enhance their training programs, which were performed inside and/or outside of their homes. Weekly telephone call was made by the physical therapist to resolve any questions.	CG: **6 weeks** standard post-PCI care protocol, involving a paper-based and self-study CHD booklet (with advice about management of their lifestyle and risk factors) and a biweekly outpatient review by clinicians.	Physical activity is monitored using remote system (belt strap with sensor, mobile app, servers, web portal). The participants wore the sensor and turned on the application on their smartphone every time they began the exercise training.	Exercise capacity (6MWT), HRQoL (SF-36), anxiety and depression (CDS score), blood pressure, and risk factors (FTND score).	6MWT, SF36, FTND, and CDS significantly improved in both groups compared with baseline. In addition, the improvements in SF36, FTND scores, and 6MWT distance were significantly higher in the IG.
Frederix et al., Belgium, 2016 [28]	2-arm RCT	IG: n = 70 CG: n = 70	CAD	IG: 12-week conventional center-based cardiac rehabilitation program, including 45 multidisciplinary rehabilitation sessions with at least two training sessions per week + **24 weeks of telerehabilitation** using both physical activity telemonitoring and dietary/smoking cessation/physical activity telecoaching strategies. Frequency: 3 times/week. Intensity: heart rate set at 70–80% heart rate reserve Time: Minimal 30 min/session. Type: Exercise protocols based on maximal CPET and calculated BMI.	CG: **12-week conventional center-based** cardiac rehabilitation program, including 45 multidisciplinary rehabilitation sessions with at least two training sessions per week + **24 weeks of conventional center-based** CR program.	Patients of IG wear the accelerometer and transmit their registered activity data weekly to the telerehabilitation center’s local server. These data enabled a semi-automatic telecoaching system to provide the patients with feedback, encouraging them to gradually achieve pre-defined exercise training goals. In addition, patients received e-mails and/or SMSs (text messages) with tailored dietary and smoking cessation recommendations.	Primary: Cost-effectiveness analysis (QALY). Secondary: Re-hospitalization rate.	The addition of cardiac telerehabilitation to conventional center-based cardiac rehabilitation is more effective and efficient than center-based cardiac rehabilitation alone.
Jozwik et al., Poland, 2021 [29]	2-arm RCT	IG: n = 50 CG: n = 50	CAD	IG: 8 sessions of standard CR (3 times per week) + **8 sessions of VR** therapy using the VR TierOne device. Standard CR Type: 40 min of interval training on a cycle ergometer Intensity: Prescribed individually based on the HRR. + 15 min break + Type: 40 min of general fitness exercises (treadmill, a pec fly machine, an elliptical trainer, a rowing machine, and a stepper) + calming phase (about 10 min) + VR therapy.	CG: 8 sessions of standard CR (3 times per week) + **8 sessions of SAT.** Standard CR. Type: 40 min of interval training on a cycle ergometer. Intensity: Prescribed individually based on the HRR. + 15 min break + Type: 40 min of general fitness exercises (treadmill, a pec fly machine, an elliptical trainer, a rowing machine, and a stepper) + Calming phase (about 10 min) + SAT played from a CD recording (for 20 min).	VR TierOne device (Stolgraf^®^, Stanowice, Poland) that comprises a computer dedicated to processing 3D graphics, VR goggles (HTC VIVE PRO, 2017, New Taipei City, Taiwan), enabling the display of high-resolution images with high picture quality (90 Hz), and manipulators that transfer the patient’s hand movements into the VR world. Therapy designed to be used with this solution is based on the metaphor of a Virtual Therapeutic Garden where the patient is able to calm down and relax.	Primary: Anxiety and depression levels (HADS). Secondary: Emotional tension level, external stress level, and intrapsychic stress level (PSQ).	In the IG, the overall HADS score was statistically significantly reduced, so VR enhanced CR for individuals with cardiovascular disease and reduced the level of anxiety and depression symptoms compared to standard CR.
Jozwik et al., Poland, 2021 [30]	2-arm RCT	IG: n = 17 CG: n = 26	CAD	IG: 8 weeks of standard CR (3 times per week) + **8 sessions of VR** therapy using the VR TierOne device. Standard CR. Type: 40 min cycle ergometer Intensity: from 60% to 85% of the maximum heart rate + Type: 40 min of general fitness exercises, with the use of a treadmill, a rowing machine, a multi-fitness station, an elliptical machine, and a cycle ergometer. The subjects exercised for 5 min using each machine, with a 2 min break in between + VR therapy.	IG: 8 weeks of standard CR (3 times per week) + **8 sessions of SAT.** Standard CR. Type: 40 min cycle ergometer Intensity: From 60% to 85% of the maximum heart rate + Type: 40 min of general fitness exercises, with the use of a treadmill, a rowing machine, a multi-fitness station, an elliptical machine, and a cycle ergometer. The subjects exercised for 5 min using each machine, with a 2 min break in between. + SAT led by a psychologist	VR TierOne device (Stolgraf^®^, Stanowice, Poland) that comprises a computer dedicated to processing 3D graphics, VR goggles (HTC VIVE PRO, 2017, New Taipei City, Taiwan), enabling the display of high-resolution images with high picture quality (90 Hz), and manipulators that transfer the patient’s hand movements into the VR world. Therapy designed to be used with this solution is based on the metaphor of a Virtual Therapeutic Garden where the patient is able to calm down and relax.	Anxiety and depression levels (HADS). Emotional tension level, external stress level, and intrapsychic stress level (PSQ).	Statistically significant differences in the efficacy of rehabilitation between groups were recorded in relation to the level of perceived stress in the sub-scales: emotional tension, external stress, intrapsychic stress, and the generalized stress scale. VR therapy is an efficient and interesting supplement to CR, with proven efficacy in reducing stress levels.
Lima et al., Brasil, 2020 [31]	2-arm RCT	IG: n = 25 CG: n = 31	Patients undergoing CABG	IG: VR group. Free active kinesiotherapy, cycle ergometry, ambulation, and re-expansive ventilatory patterns + virtual reality intervention Frequency: Twice a day for **5 days.**	CG: Free active kinesiotherapy, cycle ergometry, ambulation, and re-expansive ventilatory patterns. Frequency: Twice a day for **5 days.**	XBOX 360 platform, in addition to the Kinect electronic device, and the game used was the Kinect Sports Ultimate Collection, table tennis mode. The patients were positioned in orthostasis, facing the TV, with 20 min of daily practice time, over a period of five consecutive days, performing functional movements, such as elbow flexion–extension and internal and external rotation of the upper limbs, as well as adduction and abduction, hip dissociation, and lowering of weight in lower limbs.	Pulmonary function was assessed using MIP, MEP, VC, PEF, and functionality through the measurement of FIM and TUG.	The decrease in pulmonary variables in both pre- and post-surgery groups was noted. However, patients who participated in the virtual reality group daily, when compared to patients in the CG, demonstrated significant differences assessed on the day of hospital discharge. VR can be a tool capable of accelerating the rehabilitation process, providing greater motivation and commitment to these patients, and contributing to greater motor learning, physical capacity, and a better quality of life.
Maddison et al., New Zealand, 2015 [32]	2-arm RCT	IG: n = 85 CG: n = 86	CAD (angina; myocardial infarction; or revascularization, including angioplasty, stent, or coronary artery bypass graft within the previous 3–24 months)	IG: **24 weeks**. Patients participated in a home-based program that consisted of (1) regular exercise prescription, (2) provision of behavior change strategies, and (3) technical support. Type: Moderate to vigorous aerobic-based exercise Time: A minimum of 30 min (preferably more) most days (at least 5) of the week + personalized, automated package of text messages and a secure website with video messages aimed at increasing exercise behavior.	CG: Usual care with encouragement to be physically active and attend a cardiac club.	HEART program—a personalized, automated package of text messages via their mobile phones aimed at increasing exercise behavior over 24 weeks. They received 6 messages per week for the first 12 weeks, 5 messages per week for 6 weeks, and then 4 messages per week for the remaining 6 weeks (total 24 weeks).	Primary: Physical fitness with CPET parameter pVO2. Secondary: Self-reported physical activity (IPAQ), HRQoL with SF36 and EQ-5D, self-efficacy (task and barrier), and motivation.	A mobile phone intervention was not effective at increasing exercise capacity over and above usual care. The intervention was effective and probably cost-effective for increasing physical activity and may have the potential to augment existing cardiac rehabilitation services.
Maddison et al., New Zealand, 2019 [33]	2-arm RCT	IG: n = 82 CG: n = 80	CAD	IG: **12 weeks** of individualized exercise prescription, exercise monitoring, and coaching + theory-based behavioral strategies to promote exercise and habitual physical activity, delivered via a bespoke telerehabilitation platform. Frequency: 3 sessions per week (and encouragement to be active ≥5 days per week) Time: 30 to 60 min (including warm-up and cooldown phases) Intensity: 40–65% HRR	CG: **12 weeks** of supervised exercise delivered by clinical exercise physiologists in cardiac rehabilitation clinics. Frequency: 3 sessions per week (and encouragement to be active ≥5 days per week). Time: 30 to 60 min (including warm-up and cooldown phases). Intensity: 40–65% heart rate reserve.	REMOTE-CR platform is comprised of a smartphone and chest-worn wearable sensor (BioHarness 3, Zephyr Technology, Annapolis, MD, USA), bespoke smartphone and web apps, and custom middleware. The sensor provides information on heart and respiratory rates, single-lead ECG, and accelerometry. Smartphones with a mobile data subscription (NZD1.50/GBP0.76 per week) were loaned to participants at no cost. Custom middleware connected to smartphone and web apps. Authentication protocols in both apps, a secure web server, and encrypted data transmission ensured security and privacy.	Primary: Physical fitness with CPET parameter VO2 max. Secondary: Blood lipid and glucose concentrations, anthropometry, blood pressure, physical activity (accelerometry), exercise-related motivation, exercise adherence, adverse events (any self-reported change in health state), HRQoL with EQ-5D, program delivery, hospital service utilization, and medication costs.	VO2 max was comparable in both groups at 12 weeks, and IG was non-inferior to CG. IG participants were less sedentary at 24 weeks, while IG participants had smaller waist and hip circumferences at 12 weeks. Per capita program delivery and medication costs were lower for IG. Hospital service utilization costs were not statistically significantly different.
Piotrowicz et al., Poland, 2015 [34]	2-arm RCT	IG: n = 77 CG: n = 34	HF patients with implantable electronic devices (CIEDs).	IG: **8 weeks**. Time: 45–60 min: a warm-up lasting 5–10 min (breathing and light resistance exercises, calisthenics), a 15–45 min NW training, and a 5 min cooldown. Intensity: 40–70% of the HR reserve. Type: Aerobic NW training. Frequency: 3 days/week.	CG: **8 weeks of** usual care. Patients were not provided with a formal exercise training prescription and did not perform supervised rehabilitation.	The EHO 3 system was used to monitor and control training in any place where the patient elected to exercise. The device had training sessions preprogrammed individually for each patient (defined exercise duration, breaks, and timing of ECG recording). The planned training sessions were executed with the device indicating what should be performed with sound and light signals. The timing of automatic ECG recordings corresponded to peak exercise. If the training session was completed uneventfully, the patient transmitted the ECG recording via the mobile phone to the monitoring center immediately after the end of every training session. Patients could also transmit an ECG recording at any time, for example, if they experienced symptoms like palpitations, chest pain, etc.	Primary: Physical fitness with CPET parameter pVO2. Secondary: Workload duration (t) in CPET, 6-MWT distance and QoL with SF36, safety, and adherence to and acceptance of NW.	NW resulted in significant improvement in VO2peak, 6-MWT, and QoL. We did not observe favorable results in the CG. In neither group were there deaths or necessity for hospitalization. We did not observe any intervention from CIEDs during NW. All patients in the IG completed rehabilitation and accepted it well.
Piotrowicz et al., Poland, 2020 [35]	2-arm RCT	IG: n = 425 CG: n = 425	HF whit NYHA I–III and LVEF ≤ 40%	IG: **9-week TR program** 1 week conducted at hospital + **8-week** home-based TR 5 times/ week. **Aerobic endurance training** Devices: Nordic walking poles. Training session consists of the following: 1. Warm-up: breathing and light resistance exercises using poles for Nordic walking; duration 5–10 min. 2. Nordic walking training. Intensity: 40–70% of HRH, 11–12 on the Borg scale. Duration: Start at 10 min/session/day and gradually increase to 30–45 min/session/day. 3. Cooldown: Relaxation, breathing exercise; duration 5 min. Frequency: 1 session/day. **Respiratory muscle training** Devices: Train Air software—during the initial stage at the hospital Threshold Inspiratory Muscle Trainer—during the basic stage at home. Intensity: Start at 30% of the maximal inspiratory mouth pressure (PImax) and readjust to a maximum of 60% (if possible). Duration: Minimum 5–10 min/day; maximum 20–30 min/day. Frequency: 3–5 times/throughout the day. **Resistance and strength** **training** Devices: Thera Band—yellow color. Intensity: 5–10 repetitions of each of the seven exercises. Duration: Gradually increased 5–10–15 min/day. Frequency: 1 session/ day.	CG: Usual care. Baseline clinical examinations during a 3-day hospitalization and were then under observation until the end of the ninth week and received usual care appropriate for their clinical status and standardized within a particular center. Some of them could participate in rehabilitation, and some of them had remote monitoring of CIEDs. After the ninth week, patients underwent final assessments during a 3-day hospitalization. Patients received recommendations for suitable lifestyle changes and self-management.	The monitoring system included the following: (1) a special remote device for tele-ECG-monitored and supervised exercise training—TR set (manufactured by Pro Plus Company, Warsaw, Poland)—which consists of EHO mini device, blood pressure measuring, and weighing machine; (2) data transmission set via a mobile phone; and (3) a monitoring center capable of receiving and storing patients’ medical data (specialized hardware and software were necessary).	Primary: Percentage of days alive and out of the hospital (DAHO). Secondary: All-cause and cardiovascular mortality and all-cause cardiovascular and HF hospitalization. Tertiary: Functional capacity (change in peak VO2 and 6MWD) and QoL with SF-36.	Positive effects of 9-week hybrid comprehensive telerehabilitation on functional capacity and QoL in patients with HF do not lead to the increase in days alive and out of the hospital and do not reduce mortality and hospitalization over 14 to 26 months of follow-up.
Snoek et al., 2021, Netherlands [36]	2-arm RCT	IG: n = 61 CG: n = 61	CAD (ACS, post-PCI, or post-CABG)	IG: **After CR,** patients were randomized into **24 weeks** of training programs and equipped with a smartphone and Bluetooth-connected heart rate belt. Frequency: 5 days/week Intensity: Moderate intensity is defined as intensity above VT_1_ based on CPET. Time: At least 30 min/day. Type: Patients are free to choose type of exercise (i.e., walking, cycling).	CG: Usual care. **After CR**, patients receive a traditional **24-week** follow-up program with monthly calls by a research nurse without advice on physical activity.	Patients were equipped with a smartphone (Samsung Galaxy Ace GT-s5830i, Seul, Republic of Korea) and Bluetooth-connected heart rate belt (Zephyr, Annapolis, MD, USA). The device registered training mode, time, and intensity (determined by heart rate). Patients had insight into their training history through an individual webpage and on their smartphones. They were contacted weekly by telephone for supportive guidance in the first month, every other week in the second month, and from then on monthly until six months. Recorded physical activity patterns were discussed.	Primary: Physical fitness with CPET parameter pVO2. Secondary: QoL with Quality of Life after Myocardial Infarction Questionnaire, cardiovascular risk factors (BMI, blood pressure, total cholesterol), care usage, major adverse cardiovascular events, habitual physical activity (IPAQ), and emotional and social functioning (PHQ-9 and HADS).	Extending CR with a heart-rate-based telerehabilitation program did not yield additional sustainable health benefits compared with regular care with monthly telephone calls. Both telerehabilitation and usual care with monthly telephone calls may prevent the typically observed reductions in peak VO2 following the completion of a CR program.
Vieira et al., Portugal, 2017 [37]	3-arm RCT	IG: n = 15 IG: n = 15 CG: n = 16	CAD	IG1: Allocated to a home-based CR program using a computer and Kinect (virtual reality format). IG2: Allocated to a home-based CR program using a paper booklet (conventional format). Exercise prescription: Time: Warm up 10 min + workout strength 20–25 min + endurance 35–45 min) + stretching 6 min. Intensity: Strength −> to each individual repetition is calculated by 65–70 of the HR reserve; endurance −> to each individual repetition calculated by 65–70% of the HR reserve. The exercise protocol was performed 3 times a week over **24 weeks**. In addition, in the remaining days, a daily walk of 30 min was recommended.	CG: Only subjected to education regarding the cardiovascular risk factors.	Kinect-RehabPlay project relies on software to monitor and evaluate the rehabilitation exercises, which have to be performed by the user and captured by the Kinect sensor, providing real-time feedback. This system provides a virtual physical therapist performing the exercise and providing indications concerning the quality of execution. The participant is also represented as a second avatar, which interactively follows the physical therapist. The software uses Microsoft Kinect to track individual movements and make a match with a pre-defined pattern. This feature monitored the number of repetitions for each exercise, according to the pre-calculated value, and set it to the individual exercise profile.	Body composition (bioimpedance scale and tape measure), physical activity (accelerometer), eating patterns (Semi-Quantitative Food Frequency Questionnaire), and lipid profile (laboratory tests performed after the termination of the training phase).	The IG1 revealed significant improvements in the waist-to-hip ratio after 6 months and between the baseline and third month when compared with the CG. The IG1 also decreased their ingestion of total fat after six months and increased the high-density lipoprotein cholesterol 3 months after the program’s conclusion.
Vieira et al., Portugal, 2018 [38]	3-arm RCT	IG 1: n = 15 IG 2: n = 15 CG: n = 16	CAD	IG1: Allocated to a home-based CR program using a computer and Kinect (virtual reality format) IG2: Allocated to a home-based CR program using a paper booklet (conventional format). Exercise prescription: Time: Warm up 10 min + workout (strength 20–25 min + endurance 35–45 min) + stretching 6 min. Intensity: Strength −> to each individual repetition is calculated by 65–70 of the HR reserve; endurance −> to each individual repetition calculated by 65–70% of the HR reserve. The exercise protocol was performed 3 times a week over **24 weeks**. In addition, in the remaining days, a daily walk of 30 min was recommended.	CG: Only subjected to education regarding the cardiovascular risk factors.	Kinect-RehabPlay project relies on software to monitor and evaluate the rehabilitation exercises, which have to be performed by the user and captured by the Kinect sensor, providing real-time feedback. This system provides a virtual physical therapist performing the exercise and providing indications concerning the quality of execution. The participant is also represented as a second avatar, which interactively follows the physical therapist. The software uses Microsoft Kinect to track individual movements and make a match with a pre-defined pattern. This feature monitored the number of repetitions for each exercise, according to the pre-calculated value, and set it to the individual exercise profile.	Executive function, control, and integration in the implementation of an adequate behavior in relation to a certain objective, specifically the ability to switch information (Trail Making Test), working memory (Verbal Digit Span test), and selective attention and conflict resolution ability (Stroop test), quality of life (MacNew questionnaire), depression, anxiety, and stress (depression, anxiety, and stress scale 21).	The IG1 revealed significant improvements in the selective attention and conflict resolution ability in comparison with the CG in the variable difference and in comparison with the IG2 in the variable difference. No significant differences were found in the quality of life, depression, anxiety, and stress.

6MWT: Six-Minute Walking Test; ACS: acute coronary syndrome; CABG: coronary artery bypass grafting; CAD: coronary artery disease; CAT: COPD Assessment Test; CDS: Cardiac Depression Scale; CG: control group; CHD: coronary heart disease; CHF: chronic heart failure; COPD: chronic obstructive pulmonary disease; CPET: cardiopulmonary exercise testing; CR: cardiac rehabilitation; EF: ejection fraction; EQ-5D: EuroQol 5-dimension; FIM: functional independence; FTND: Fagerstrom Test for Nicotine Dependence; HADS: Hospital Anxiety and Depression Scale; HbA1c: glycated hemoglobin A1c; HRQoL: health-related quality of life; HRR: heart rate reserve; IG: intervention group; IPAQ: International Physical Activity Questionnaire; LDL-c: low-density lipoprotein cholesterol; MEP: maximum expiratory pressure; MET: metabolic equivalent of task; MI: myocardial infarction; MIP: maximum inspiratory pressure; MLHFQ: Minnesota Living with Heart Failure Questionnaire; MRC: Medical Research Council; NT: Nursing Tutor; NW: Nordic Walking; PASE: Physical Activity Scale for Elderly; PCI: percutaneous coronary intervention; PEF: peak expiratory flow; PHQ-9: patient health questionnaire; PSQ Perception of Stress Questionnaire; PT: Physiotherapist Tutor; QALYs: quality-adjusted life years; QoL: quality of life; RCT: randomized controlled trial; SAT: Schultz Autogenic Training; SF-36: short form—36; TUG: timed up and go; VC: vital capacity; VO2 max: maximal oxygen uptake; VO2 peak: peak oxygen consumption; VR: virtual reality; WC: waist circumference.

**Table 4 healthcare-12-01534-t004:** Results of the studies and the qualitative assessment with the PEDro scale.

Author, Country, Year	Study Design	Population, n	Diagnosis	Results	PEDro Score and Quality
Batalik et al., Czech Republic, 2020 [23]	2-arm RCT	IG: n = 28 CG: n = 28	CAD	pVO2 showed significant improvement (*p* < 0.001) in CG group from 23.4 ± 3.3 to 25.9 ± 4.1 mL/kg/min and (*p* < 0.01) in IG group from 23.7 ± 4.1 to 26.5 ± 5.7mL/kg/min. The training adherence between groups was similar.	5 (POOR)
Bernocchi et al., Italy, 2018 [24]	2-arm RCT	IG: n = 56; CG: n = 56	COPD and CHF	IG patients were able to walk further than at baseline: mean (95% CI) Δ6MWT was 60 (22.2, 97.8) m; the CG showed no significant improvement. In IG, the media time to hospitalization/death was 113.4 days compared with 104.7 in the CG (*p* = 0.0484, log-rank test). Other secondary outcomes: MRC (*p* = 0.0500), PASE (*p* = 0.0015), Barthel (*p* = 0.0006), MLHFQ (*p* = 0.0007), and CAT (*p* = 0.0000) were significantly improved in the IG compared with the CG at 4 months. IG maintained the benefits acquired at 6 months for outcomes.	6 (GOOD)
Bravo-Escobar et al., Spain, 2017 [25]	2-arm RCT	IG: n = 14 CG: n = 14	CAD	No significant differences were observed between CG and IG. For exercise time and METS achieved during the exertion test, the recovery rate in the first minute increased in both groups after the intervention. The only difference between the two groups was for quality-of-life scores (10.93 [IC95%: 17.251, 3.334, *p* = 0.007] vs. −4.314 [IC 95%: −11.414, 2.787; *p* = 0.206]).	5 (POOR)
Cai et al., China, 2022 [26]	2-arm RCT	IG: *n* = 50CG: *n* = 50	Patients who underwent ablation for atrial fibrillation	The mean VO2peak increased significantly in IG (baseline vs. 12 weeks: 19.1 ± 4.7 vs. 27.3 ± 5.6 mL/(min kg), *p* < 0.01) and the CG (baseline vs. 12 weeks: 18.7 ± 4.9 vs. 22.9 ± 6.3 mL/(min kg), *p* < 0.01). Between-group analysis of aerobic capacity was significantly in favor of the IG. During the 12-week program, patients in the IG exhibited better adherence than those in the CG. Self-reported physical activity improved more in the IG than in the CG (all *p* < 0.01).	6 (POOR)
Fang et al., China, 2019 [27]	2-arm RCT	IG: n = 40 CG: n = 40	Patients with CAD post-PCI	6MWT distance was significantly improved in both groups compared with baseline (IT 48.20 vs. CG 34.77). SF36 distance was significantly improved in both groups compared with baseline (IT 14.18 vs. CG 6.75). FTND distance was significantly improved in both groups compared with baseline (IG −2.09 vs. CG −1.06).	4 (POOR)
Frederix et al., Belgium, 2016 [28]	2-arm RCT	IG: n = 70 CG: n = 70	CAD	The total average cost per patient was significantly lower in the IG (EUR 2156 ± EUR 126) than in the CG (EUR 2720 ± EUR 276) (*p* = 0.01). The number of days lost due to cardiovascular re-hospitalizations in the IG (0.33 ± 0.15) was significantly lower than in the CG (0.79 ± 0.20) (*p* = 0.037).	7 (GOOD)
Jozwik et al., Poland, 2021 [29]	2-arm RCT	IG: n = 50 CG: n = 50	CAD	In the IG, HADS score was statistically significantly reduced by 13.5%, HADS-Depression by 20.8%, and the general stress level by 12.8% (*p* < 0.05). In the CG, the scores for the HADS, HADS-Anxiety, and general stress levels were statistically significantly higher, at 4.8%, 6.5%, and 4.9%, respectively. VR enhanced CR for individuals with cardiovascular disease reduced the level of anxiety and depression symptoms compared to standard CR.	5 (POOR)
Jozwik et al., Poland, 2021 [30]	2-arm RCT	IG: n = 17 CG: n = 26	CAD	In the CG, there was a deterioration in nearly all tested parameters except for HADS-Depression. Statistically significant differences in the efficacy of rehabilitation were recorded in relation to the level of stress in the sub-scales: emotional tension (*p* = 0.005), external stress (*p* = 0.012), intrapsychic stress (*p* = 0.023), and generalized stress scale (*p* = 0.004).	5 (POOR)
Lima et al., Brasil, 2020 [31]	2-arm RCT	IG: n = 25 CG: n = 31	Patients undergoing CABG	MIP of the CG was 74 ± 15 vs. 92 ± 12 cmH_2_O of the IG (*p* < 0.001). MEP of the GC was 54 ± 14 vs. 75 ± 16 cmH_2_O of the IG (*p* < 0.001). VC was 1.9 ± 0.6 mL/Kg in GC vs. 2.4 ± 0.7 mL/Kg in IG (*p* = 0.22). PEF in GC was 231 ± 28 vs. 311 ± 26 L/min in IG (*p* < 0.001). TUG of CG 22 ± 9.1 s vs. 10 ± 1.6 s in the IG (*p* < 0.001). FIM of CG was 112 ± 5 vs. 120 ± 3 in the IG (*p* < 0.001). VR was effective in reducing the loss of pulmonary function and functional independence after CABG.	5 (POOR)
Maddison et al., New Zealand, 2015 [32]	2-arm RCT	IG: n = 85 CG: n = 86	CAD	No differences in PVO2 between the two groups (difference—0.21 mL kg^−1^ min^−1^, 95% CI: −1.1, 0.7; *p* = 0.65) at 24 weeks. Significant treatment effects were observed for selected secondary outcomes, including leisure time physical activity (difference 110.2 min/week, 95% CI: −0.8, 221.3; *p* = 0.05) and walking (difference 151.4 min/week, 95% CI: 27.6, 275.2; *p* = 0.02). Significant improvements in self-efficacy to be active (difference 6.2%, 95% CI: 0.2, 12.2; *p* = 0.04) and the general health domain of the SF36 (difference 2.1, 95% CI: 0.1, 4.1; *p* = 0.03) at 24 weeks.	8 (GOOD)
Maddison et al., New Zealand, 2019 [33]	2-arm RCT	IG: n = 82 CG: n = 80	CAD	VO2 max was comparable in both groups at 12 weeks, and IG was non-inferior to CG-adjusted mean difference = 0.51 (95% CI −0.97 to 1.98) mL/kg/min, *p* = 0.48). IG participants were less sedentary at 24 weeks (AMD = −61.5 (95% CI −117.8 to −5.3) min/day, *p* = 0.03). CG participants had smaller waist AMD = 1.71 (95% CI 0.09 to 3.34) cm, *p* = 0.04, and hip circumferences AMD = 1.16 (95% CI 0.06 to 2.27) cm, *p* = 0.04 at 12 weeks.	8 (GOOD)
Piotrowicz et al., Poland, 2015 [34]	2-arm RCT	IG: n = 77 CG: n = 34	HF patients with implantable electronic devices (CIEDs)	IG resulted in significant improvement in VO2 peak (16.1 ± 4.0 vs. 18.4 ± 4.1 (mL/kg/min), *p* = 0.0001), t (471 ± 141 vs. 577 ± 158 (s), *p* = 0.0001), 6-MWT (428 ± 93 vs. 480 ± 87 (m), *p* = 0.0001), and QoL (79.0 ± 31.3 vs. 70.8 ± 30.3 (score), *p* = 0.0001). No favorable results in the CG.	5 (POOR)
Piotrowicz et al., Poland, 2020 [35]	2- arm RCT	IG: n = 425 CG: n = 425	HF whit NYHA I–III and LVEF ≤ 40%	The IG did not extend the percentage of days alive and out of the hospital; the mean days were 91.9 (19.3) days in the IG vs. 92.8 (18.3) days in the CG. During follow-up, 54 patients died in the IG and 52 in the CG, with mortality rates at 26 months of 12.5% vs. 12.4%, respectively (hazard ratio, 1.03 [95% CI, 0.70–1.51]). There were also no differences in hospitalization rates (hazard ratio, 0.94 [95% CI, 0.79–1.13]). The IG significantly improved pVO2 (0.95 [95% CI, 0.65–1.26] mL/kg/min vs. 0.00 [95% CI, −0.31 to 0.30] mL/kg/min; *p* < 0.001) and quality of life, 1.58 [95% CI, 0.74–2.42] vs. 0.00 [95% CI, −0.84 to 0.84]; *p* = 0.008).	8 (GOOD)
Snoek et al., 2021, Netherlands [36]	2-arm RCT	IG: n = 61 CG: n = 61	CAD	PeakVO2 increased significantly from baseline to 12 months in IG (+2.5 mL kg^−1^min^−1^ (95% CI 1.5–3.2)) and CON (+1.9 mL kg^−1^min^−1^ (95% CI 1.0–2.5)), and did not differ between groups (*p* = 0.28). QoL (*p* = 0.31), total cholesterol (*p* = 0.45), and major adverse cardiovascular events (*p* = 0.86) did not differ between groups and in time.	6 (GOOD)
Vieira et al., Portugal, 2017	3-arm RCT	IG: n = 15 IG: n = 15 CG: n = 16	CAD	IG 1 revealed significant improvements in the waist-to-hip ratio after 6 months (*p* = 0.033) and between the baseline and third month when compared with the CG (*p* = 0.041). IG1 decreased their ingestion of total fat (*p* = 0.032) after 6 months and increased the high-density lipoprotein cholesterol (*p* = 0.017) 3 months after the program’s conclusion.	5 (POOR)
Vieira et al., Portugal, 2018	3-arm RCT	IG 1: n = 15 IG 2: n = 15 CG: n = 16	CAD	IG1 revealed significant improvements in the selective attention and conflict resolution ability in comparison with the CG in the variable difference M0–M2 (*p* = 0.021) and in comparison with the IG2 in the variable difference M1–M2 and M0–M2 (*p* = 0.001 and *p* = 0.002, respectively). No significant differences were found in the quality of life, depression, anxiety, and stress.	5 (POOR)

6MWT: Six-Minute Walking Test; CABG: coronary artery bypass grafting; CAD: coronary artery disease; CAT: COPD Assessment Test; CDS Cardiac Depression Scale; CG: control group; CI: confidence interval; CHF: chronic heart failure; COPD: chronic obstructive pulmonary disease; CPET: cardiopulmonary exercise testing; CR: cardiac rehabilitation; FIM: functional independence; HADS: Hospital Anxiety and Depression Scale; IG: intervention group; MEP: maximum expiratory pressure; MET: metabolic equivalent of task; MI: myocardial infarction; MIP: maximum inspiratory pressure; MRC: Medical Research Council; PASE: Physical Activity Scale for Elderly; PCI: percutaneous coronary intervention; PEF: peak expiratory flow; QoL: quality of life; RCT: randomized controlled trial; SF-36: short form—36; TUG: timed up and go; VC: vital capacity; VO2 max: maximal oxygen uptake; VO2 peak: peak oxygen consumption; VR: virtual reality.

## Data Availability

All data generated or analyzed during this study are included in this published article.

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
