# Peer review of "Technological Developments, Exercise Training Programs, and Clinical Outcomes in Cardiac Telerehabilitation in the Last Ten Years: A Systematic Review"

_healthcare, 2024, doi:10.3390/healthcare12151534_

Round 1

Reviewer 1 Report

Comments and Suggestions for Authors

This systematic review addresses the topic of telerehabilitation for patients with cardiovascular diseases in a detailed and well-explained manner. Below are comments and suggestions to improve the quality of the paper:

Introduction: Provide a more detailed description of traditional Cardiac Rehabilitation in the introduction to clarify what is encompassed by this term.

Methodology: The authors mentioned using PubMed, Scopus, and Web of Science. Explain why other relevant databases were not considered.

Results: Table 3 on study characteristics is excellent and comprehensive. The authors should consider summarizing the results of all analyzed studies in a table or another format to make it clearer to readers what each study found. Additionally, this table could include PEDro scores for each study to simultaneously show the study quality and its results.

Discussion: Are there any studies that describe patient compliance with telerehabilitation in general? Mention them in the discussion if they exist.

Conclusion: The conclusion should also address potential suggestions for future research that would investigate long-term benefits.

References: There are inconsistencies in the citations; for example, reference 34.

Author Response

Comments 1: Introduction: Provide a more detailed description of traditional Cardiac Rehabilitation in the introduction to clarify what is encompassed by this term.

Response 1: Thanks for the observation, we have proceeded to modify. Cardiac Rehabilitation (CR), that is an active and dynamic multifactorial process that includes exercise training, cardiac risk factor modification, psychosocial assessment, and outcomes assessment [4], which have demonstrated favorable effects on clinical stability, on reducing disability and the risk of subsequent cardiovascular events, on supporting the maintenance and resumption of an active role in society, and on improving survival and quality of life [5].

Comments 2: Methodology: The authors mentioned using PubMed, Scopus, and Web of Science. Explain why other relevant databases were not considered.

Response 2: Upon preliminary analysis of the other databases, the results were comparable, so we decided to use the databases PubMed, Scopus, and Web of Science, which are representative for rehabilitation, however, following your comment, we decided to report this choice within the limitations of the study.

Comments 3: Results: Table 3 on study characteristics is excellent and comprehensive. The authors should consider summarizing the results of all analyzed studies in a table or another format to make it clearer to readers what each study found. Additionally, this table could include PEDro scores for each study to simultaneously show the study quality and its results.

Response 3: thank you for your suggestion, we have included the table in the text

Comments 4: Discussion: Are there any studies that describe patient compliance with telerehabilitation in general? Mention them in the discussion if they exist.

Response 4: thank you for your suggestion, there are studies on patient compliance with telerehabilitation, so we have included the changes in the text

Comments 5: Conclusion: The conclusion should also address potential suggestions for future research that would investigate long-term benefits.

Response 5: thank you for your suggestion, we have included the changes in the text

Comments 6: References: There are inconsistencies in the citations; for example, reference 34.

Responce 6: thank you for your report, we have proceeded to edit

Reviewer 2 Report

Comments and Suggestions for Authors

Dear Editor,

The following suggestions are made to improve the quality of the study.

  • It is better to discuss the global consequences of cardiovascular disease and the need to use telemedicine technology in this field in the introduction section to further explain the necessity of conducting the present study.
  • According to the study design, is the current research not registered in databases such as PROSPERO or similar?
  • Why did the authors limit themselves to three databases and did not search databases such as Cochrane Library and Embase? This can be a limitation of the study.
  • It is not necessary to state the exclusion criteria that are the reverse of the inclusion criteria. Because they are somehow mentioned in the inclusion criteria section, it is necessary to modify the inclusion and exclusion criteria.
  • Why was it necessary for the authors to evaluate the 16 eligible articles based on the PEDro Score? Because it was not intended to discard an article based on the method mentioned above in case of a low score, and only the quality of the articles was checked.
  • At the end of the method section, it is expected that there will be a sub-section under the title of "data analysis" or "data extraction" for the method of data extraction or analysis to determine how the data extracted from the articles will be summarized or analyzed.
  • In figure 1, indicate how many articles were excluded based on title and abstract, respectively.
  • In the conclusion section, the authors are expected to draw conclusions based on the sum of available findings and summarize them. Therefore, referring to two sources numbers 8 and 9 is not an interesting result and the authors should draw a general conclusion.

Author Response

Comments 1: It is better to discuss the global consequences of cardiovascular disease and the need to use telemedicine technology in this field in the introduction section to further explain the necessity of conducting the present study.

Response 1: thank you for your report, we have proceeded to edit

Comments 2: According to the study design, is the current research not registered in databases such as PROSPERO or similar?

Response 2: No, we did not registered our review

Comments 3: Why did the authors limit themselves to three databases and did not search databases such as Cochrane Library and Embase? This can be a limitation of the study.

Response 3: Upon preliminary analysis of the other databases, the results were comparable, so we decided to use the databases PubMed, Scopus, and Web of Science, which are representative for rehabilitation, however, following your comment, we decided to report this choice within the limitations of the study.

Comments 4: It is not necessary to state the exclusion criteria that are the reverse of the inclusion criteria. Because they are somehow mentioned in the inclusion criteria section, it is necessary to modify the inclusion and exclusion criteria.

Response 4: thank you for your report, we have proceeded to edit

Comments 5: Why was it necessary for the authors to evaluate the 16 eligible articles based on the PEDro Score? Because it was not intended to discard an article based on the method mentioned above in case of a low score, and only the quality of the articles was checked.

Response 5: We decided to use the Pedro scale, but not to use the score as an exclusion criterion, to give the reader an additional tool for evaluating the results

Comments 6: At the end of the method section, it is expected that there will be a sub-section under the title of "data analysis" or "data extraction" for the method of data extraction or analysis to determine how the data extracted from the articles will be summarized or analyzed.

Response 6: thanks for the observation, we have proceeded to modify

Comments 7: In figure 1, indicate how many articles were excluded based on title and abstract, respectively.

Response 7: thanks for the observation, we have proceeded to modify

Comments 8: In the conclusion section, the authors are expected to draw conclusions based on the sum of available findings and summarize them. Therefore, referring to two sources numbers 8 and 9 is not an interesting result and the authors should draw a general conclusion.

Response 8: thanks for the observation, we have proceeded to modify